# Anticipating the Next Chess Move: Blocking SARS-CoV-2 Replication and Simultaneously Disarming Viral Escape Mechanisms

**DOI:** 10.3390/genes13112147

**Published:** 2022-11-18

**Authors:** Samir Mansour Moraes Casseb, André Salim Khayat, Jorge Estefano Santana de Souza, Edivaldo Herculano Correa de Oliveira, Sidney Emanuel Batista Dos Santos, Pedro Fernando da Costa Vasconcelos, Paulo Pimentel de Assumpção

**Affiliations:** 1Oncology Research Center, Federal University of Pará, Belém 66073-000, Brazil; 2Bioinformatics Graduate Program, Metrópole Digital Institute, Federal University of Rio Grande do Norte Natal, Natal 59078-400, Brazil; 3Department of Pathology, Pará State University, Belém 66087-670, Brazil

**Keywords:** COVID-19, SARS-CoV-2, siRNA, treatment

## Abstract

The COVID-19 pandemic initiated a race to determine the best measures to control the disease and to save as many people as possible. Efforts to implement social distancing, the use of masks, and massive vaccination programs turned out to be essential in reducing the devastating effects of the pandemic. Nevertheless, the high mutation rates of SARS-CoV-2 challenge the vaccination strategy and maintain the threat of new outbreaks due to the risk of infection surges and even lethal variations able to resist the effects of vaccines and upset the balance. Most of the new therapies tested against SARS-CoV-2 came from already available formulations developed to treat other diseases, so they were not specifically developed for SARS-CoV-2. In parallel, the knowledge produced regarding the molecular mechanisms involved in this disease was vast due to massive efforts worldwide. Taking advantage of such a vast molecular understanding of virus genomes and disease mechanisms, a targeted molecular therapy based on siRNA specifically developed to reach exclusive SARS-CoV-2 genomic sequences was tested in a non-transformed human cell model. Since coronavirus can escape from siRNA by producing siRNA inhibitors, a complex strategy to simultaneously strike both the viral infectious mechanism and the capability of evading siRNA therapy was developed. The combined administration of the chosen produced siRNA proved to be highly effective in successfully reducing viral load and keeping virus replication under control, even after many days of treatment, unlike the combinations of siRNAs lacking this anti-anti-siRNA capability. Additionally, the developed therapy did not harm the normal cells, which was demonstrated because, instead of testing the siRNA in nonhuman cells or in transformed human cells, a non-transformed human thyroid cell was specifically chosen for the experiment. The proposed siRNA combination could reduce the viral load and allow the cellular recovery, presenting a potential innovation for consideration as an additional strategy to counter or cope COVID-19.

## 1. Introduction

Severe acute respiratory syndrome coronavirus 2 (SARS-CoV-2) is the etiologic agent of COVID-19 (Coronavirus Disease, 2019), the cause of the pandemic. The SARS-CoV-2 is a 30 kb positive single-stranded RNA virus with 10 open reading frames (ORFs) that encode for structural, non-structural, and accessory proteins. The second-largest open reading frame of SARS-CoV-2 is called ORF1ab, and it encodes the replicase polyprotein 1ab (7096 amino acids). Polyprotein 1ab must be processed by the virally encoded proteases chymotrypsin-like protease (3CLpro, NSP5) and papain-like protease (PAPP) (PLpro, NSP3). Both proteases were initially a part of Polyprotein 1ab before being auto catalytically released. The proteases then cleave the remaining Polyprotein 1ab proteins into a total of 16 non-structural proteins that are involved in viral genome replication and transcription [1,2]. The remainder (10 kb) of the genome encodes five structural proteins, including nucleocapsid (N), membrane (M), surface (S), and envelope (E) proteins, as well as accessory proteins, including ORF3, ORF6, ORF7, ORF8, and ORF10 [1,2,3].

Based on the sequence similarity, the virus was initially identified in Wuhan, China, and believed to have originated from BatCov RaTG13. Individuals infected with COVID-19 may experience a variety of symptoms, including fever or chills, cough, fatigue, muscle aches, headache, or diarrhea. The NIH has classified COVID-19 into five distinct types, including asymptomatic, mild, moderate, severe, and critical illness, based on the severity of the symptoms. Intensive care and intubation may be required for patients with severe respiratory illness or acute respiratory distress syndrome, and this frequently results in death. It appears that age and the presence of underlying comorbidities determine the course and outcome of diseases [4,5].

Global efforts to control COVID-19 resulted in a race for potentially efficient therapies against SARS-CoV-2; Despite efforts, few drugs have an efficient treatment for COVID-19, we can mention Paxlovid as one of these initiatives, which is an active inhibitor of 3Cl protease as a treatment demonstrating some efficiency [6,7].

The speed of vaccine development conflicts with the speed of virus mutations that enable the permanency of very high indices of infections and deaths and decrease vaccine efficacy over time, making disease control a major challenge [8,9,10].

Although the combination of population vaccination and protective measures such as facemasks, hygienic protocols, and social distancing greatly contributed to COVID-19 control, the world is still at risk of losing this fragile equilibrium due to the threat of new epidemics [3,11].

Most of the initiatives to find a putative treatment against SARS-CoV-2 are based on testing many molecules and drugs used for other diseases, aiming to discover a secondary benefit against COVID-19. Unfortunately, no drug was found to present or support an efficient and specific treatment [10,12].

Although vaccines and protective measures have saved millions of lives, efficient and real control of COVID-19 was not reached. Intriguingly, each great scientific advance for controlling the pandemic was countered by the emergence of more aggressive and/or infectious variants of SARS-CoV-2 that changed the game in favor of virus success [13].

Small interfering RNAs (RNAi) were discovered in 1998 and, over time, have become a piece with great potential for developing therapeutic drugs. RNAi is a conserved post-transcriptional silencing pathway that can regulate gene expression [14].

Small complementary non-coding RNAs can modulate messenger RNA (mRNA) translation and its stability during RNAi action. Among the RNAi, the most popular for use in therapies are the small interfering RNAs (siRNAs), and these RNAs may have an antiviral function in nature and thus protect plants, fungi, and invertebrates from viral attacks [15].

The endogenous RNAi pathway occurs in two phases: during the first one, the so-called initiation phase, the long dsRNA is cleaved by Dicer endoribonuclease into siRNAs. This generated short double-stranded dsRNAs (21–23 bp) with a sequence of 19 nucleotides complementary to the target mRNA. Then, the siRNA strands unwind and, in the second phase, known as the effector phase, the guide or antisense strand of the siRNAs is loaded into the multiprotein RNA-induced silencing complex (RISC) that further guides the RISC to recognize and cleave the target transcript, causing gene silencing, it is noteworthy that this action is catalyzed by AGO2 [16].

For many viruses or strains, there are no available antivirals, and current antivirals are limited by toxicity and drug resistance. Consequently, alternative methods, such as RNA interference (RNAi), are necessary. RNA interference inhibits the expression of any mRNA, making it a promising candidate for antiviral therapies. Several studies evaluating siRNAs in a variety of viruses have yielded promising results. However, stability and delivery issues with siRNAs persist. By modifying the structure of the siRNA, employing an efficient delivery vector, and targeting multiple regions of the viral genome with a single dose, these issues can be mitigated. Finding these answers could expedite the development of RNAi-based antivirals [11,12,13].

Notwithstanding the above, it is worthwhile to generate siRNA targets, as there are a number of variables to consider. When this difficulty is overcome, the approach will become extremely promising once viral-specific targets are designed [14].

Since SARS-CoV-2 is an RNA virus, promising initiatives using siRNAs to block the replication of coronavirus, targeted antiviral therapy using siRNAs should be attempted, in addition to the current strategy of providing vaccination and other protection measures to control COVID-19 [17,18].

Aiming to selectively impair viral replication and simultaneously anticipate viral reactions, including an siRNA escape mechanism, as previously described, i.e., preventing a SARS-CoV-2 escape strategy that could reduce the sustained therapeutic response, a combined targeted therapy using siRNAs directed to both combat the virus and its siRNA escape mechanism was developed and tested in a cellular model of non-transformed human cells. The proposed strategy has proven to be safe and extremely efficient in consistently reducing viral load while reestablishing cellular viability.

## 2. Materials and Methods

### 2.1. Virus Isolate

The SARS-CoV-2 strain used in the experiments was taken from the sample bank belonging to the Instituto Evandro Chagas. To confirm the case of SARS-CoV-2 infection, the previously described RT–qPCR [19] was performed using the commercial kit GoTaq Probe 1-Step RT–qPCR (Promega, Madison, WI, USA).

For the isolation of positive samples, the cell culture was performed using VERO E6 cells as described elsewhere [20]. After cultivation into Vero cells and when 75% of cells showed cytopathic effects (CPEs), the culture was harvested and centrifuged at 1000× *g* for 5 min, and the supernatants were stored in aliquots at −80 °C until use.

### 2.2. The Nontransformed In Vitro Model

Nontransformed human thyroid cells (IMR-90) were cultured in DMEM as described elsewhere [21]. An MOI of 0.5 of the SARS-CoV-2 strain BIO01/2020 was used to infect the cells. The culture was maintained for 120 h post-infection (hpi), thus proving to be a good model for the siRNA experiment. This same infection protocol was used for all experiments. Each of the experiments was performed at least three times to ensure the reliability of our results.

### 2.3. Viral Load

The method described by [22] was used to quantify the viral load. Briefly, 24-well plates seeded with VERO E6 cells at a concentration of 1 × 10^4^ cells/well were used. After 24 h and 80–90% confluence, dilutions from 10^−1^ to 10^−10^ in DMEM containing 2.5% fetal bovine serum (FBS) of the virus were transferred in triplicate (100 µL/well) to seeded plates. After 1 h of adsorption at 37 °C in a 5% CO_2_ atmosphere, the wells were completed with an overlay of carboxymethylcellulose (CMC) with DMEM, 2% FBS and 1% penicillin–streptomycin mix, and the plates were incubated at 37 °C in a 5% CO_2_ incubator and stained with methylene blue dissolved in sodium acetate-acetic acid. Plates were stained after 96 hpi. Finally, the viral load in PFU/mL of each sample was determined.

### 2.4. Treatment with siRNA

A total of four siRNAs (siRNAs I, II, III and IV) targeting different sequences in the 1a/1ab ORF regions were used both individually and together. A negative control (SC2_NC) was also applied, as recommended by the producers, ensuring that the siRNA system was working properly. These siRNAs were mixed with the RNAiMax Lipofectamine delivery vehicle (Thermo Fisher Scientific, Waltham, MA, USA) at a concentration of 500 ng/well of Lipofectamine for every 5 pmol/well of siRNA, and the same concentrations were used for the SARS-CoV-2 uninfected controls.

The Silencer Select GAPDH siRNA (Ambion, Austin, TX, USA) and Silencer Select Negative Control (Ambion) systems were also applied as quality controls.

In addition, an extra siRNA, the anti-escape siRNA (siRNA-V), was combined with siRNA-I plus siRNA-III for the additional experiments, keeping the same described protocols except regarding the number of treatments, since instead of giving daily doses, the combinations of both siRNA-I plus siRNA-III and siRNA-I plus siRNA-III plus siRNA-V were applied exclusively 24 hpi as unique doses.

The regions of the siRNA used were chosen based on the database described by Medeiros et al. [18], where the author describes and evaluates possible siRNA sequences through more than 170 resources, including thermodynamic information, background context, target genes, and sequence alignment information against the human genome and various strains of SARS-CoV-2, to evaluate possible links to off-target sequences.

The siRNAs were made to be siRNAs I–IV in the ORF1ab region of SARS-CoV-2. This made sure that they could directly stop replication before the genome was cut, which would result in fewer reads and make the siRNAs less effective (Figure 1). In addition, the anti-escape siRNA was designed in the N region of SARS-CoV-2.

It is worth noting that the areas targeted for our siRNA therapy are not employed in previous research that are using siRNA as an antiviral. Furthermore, none of the other studies disclose the use of an anti-escape siRNA like ours.

### 2.5. Experimental Design

All experiments were performed at least three times to ensure data reliability. An interval of 24 h between siRNA treatments for the removal of both the cell supernatant for viral quantification and the cells for the proposed analysis was adopted (Figure A1).

### 2.6. Analysis of Cytotoxicity, Cell Viability and Cytopathic Effect

After infecting the cells with the SARS-CoV-2 strain, the analysis of cytotoxicity, caspase 3 and caspase 7 signaling and cell viability was performed using the ApoTox-Glo Triplex Assay (Promega, Madison, WI, USA); the induced cytopathic effects (CPEs) were quantified by a Viral ToxGlo Assay (Promega, Madison, WI, USA). Mitochondrial activity was also analyzed using the Mitochondrial ToxGlo Assay kit (Promega, Madison, WI, USA). All kits were utilized on the glomax multi+ platform (Promega, Madison, WI, USA) according to the instructions provided by the manufacturer.

### 2.7. IMR-90 Cell Infection, Viral Loads and Cell Viability

An MOI of 0.5 was used to infect IMR-90 cells to maintain both SARS-CoV-2 replication and cell viability over 120 hpi, allowing culture observation and viral load estimation (Appendix A).

Since the cells needed to remain viable for at least five days after infection to allow the in vitro treatment experiment, the measurements were quantified over five days, and the results are shown in Appendix B Figure A2C, confirming that the model works properly. Even with an evident decay of cell number occurring at 48 hpi, most cells remained viable until 120 hpi.

### 2.8. The Cytopathic Effect Demonstration

We used the glomax multi+ platform (Promega, USA) with the commercial kits Viral ToxGlo Assay (Promega, USA), CellTiter-Glo^®^ 2.0 Cell Viability Assay (Promega, Madison, WI, USA) to demonstrate the cytopathic effect in our test (Promega, USA). All used and according to the manufacturer’s instructions.

The SARS-CoV-2 culture showed evident morphologic CPEs (Appendix A Figure A1). In addition, ATP production was also quantified using the Glo-max platform, since the level of ATP production is directly related to CPEs (Appendix C Figure A3B). The expression of caspases 3 and 7 (Appendix C Figure A3B) and mitochondrial viability (Appendix C Figure A3A) were analyzed, and a deterioration of cellular homeostasis was demonstrated.

Finally, the relationship between viral load, cell viability and CPEs, as well as between viral load and caspase expression (Appendix C), demonstrated a deterioration of cellular homeostasis related to the increase in viral load.

Since the established model proved to be useful for the designed siRNA experiment, the next steps included the evaluation of the siRNA’s toxicit y to the cells, mainly regarding any off-target effect, to warrant strategy safety (Appendix D).

### 2.9. Statistical Analysis

The R-project program (r-project.org/, accessed on 27 September 2022) and the JAMOVI program v. 1.6 (jamovi.org, accessed on 27 September 2022) were used. ANOVA and Student’s t tests were used for group comparisons. A significance value of *p* < 0.05 and a 95% confidence interval were adopted.

## 3. Results

### siRNA Treatments

Initially, each siRNA was tested individually to verify the differences in viral load throughout the treatment and the efficacy of each siRNA (Figure 2). In addition to the viral load, cell viability was also verified throughout the experimental treatment. Next, a “cocktail” was created using all four siRNAs together. The siRNA combination increased treatment effectiveness when compared to each individually tested siRNA (Figure 2E). This test demonstrates that the combination becomes more efficient, increasing efficiency by 35% compared to siRNAs used separately.

Nevertheless, this improvement in the efficacy due to using the siRNA combination, both individual siRNA treatments, and the siRNA combination including the four siRNAs were highly efficient in provoking a marked decrease in viral load just before starting treatment and during the first three days but allowed a subtle regrowth of viral load observed at the fourth day, suggesting a deterioration in efficacy.

Addressing this hypothesis, an additional siRNA, siRNA-V, was incorporated into the experiments to block the hypothetical escape strategy of the virus.

The addition of siRNA-V prevents virus regrowth, keeping the viral load down throughout the whole experimental time. Interestingly, the inclusion of siRNA-V also improved cellular viability, strongly demonstrating the inhibition of the virus’ escape mechanism A comparison of treatments using the combination of siRNA-I plus siRNA-III, the most individually efficient siRNAs, versus the combination of siRNA-I plus siRNA-III plus the addition of siRNA-V, the anti-escape siRNAs, is demonstrated (Figure 3).

## 4. Discussion

The emergence and continuation of the SARS-CoV-2 pandemic became a challenge for healthcare professionals to rapidly combat the spread of COVID-19. Effective treatments against COVID-19 are still scarce and without widespread availability, thus becoming a major public health problem [23]. The development of an efficient, short course, and widely available treatment strategy to prevent viral replication and disease progression represents a potential weapon to control the pandemic, together with available protective measures, vaccination and medical care [17,24].

Vaccination against RNA viruses remains a challenge due to the higher rates of errors related to the virus RNA polymerases, differently to that of DNA viruses, able to self-correct most of these errors. The main clinical consequences of these “mutational burden” are lost of vaccination efficacy and recurrent outbreaks, due to the new variants’ ability to escape from available vaccines.

Regarding COVID-19, nevertheless the great improvement in worldwide production, distribution, and application of vaccines, the high mutational burden of the SARS-CoV-2 and the respiratory rout of this infection, represent a constant threat to the current stability of the disease. The replacement of virus variants for new ones repeatedly puts researchers, health providers and the whole population, in alarm.

The anxiety for having a specific and highly efficient treatment, able to block the infection, and impact transmission rates, moved many researchers to investigate innovative strategies to face COVID-19.

Among those initiatives, the siRNAs molecular therapy surge as a promising alternative, due to their high specificity and great potential to control virus replication.

Attempts using siRNA against viral infections have already been described in cases of human immunodeficiency virus, influenza virus, coxsackie B3, adenovirus, hepatitis B virus, hepatitis C virus, and to tackle SARS-CoV in various cellular and animal models [25], thus supporting the investigation of such a strategy to combat COVID-19 [17,25,26].

It is worth noting that small interfering RNAs (siRNAs) have the potential to play a crucial role in preventing the expression of disease-causing viral genes through the hybridization and subsequent inactivation of disease-causing viral mRNAs target complementarities via RNA interference (RNAi) [27].

Consequently, a targeted therapy against SARS-CoV-2 using siRNAs was proposed and proved to be highly effective in reducing the viral load of infected non-transformed thyroid human cells and, mainly, safe, since those cells were not injured by the therapy and tended to recover from viral aggression, thus opening a path to proceeding to animal and human trials.

However, it is important to note that, thus far, siRNA treatments utilize only a single target of the viral genome, i.e., one siRNA at a time. What can become a rather limited method with the potential to lose effectiveness over time and with the potential for viral genomes to undergo genetic mutations.

Aiming to achieve reliable siRNA therapy, the viral genome and human infection processes were studied to determine the best putative targets to avoid effective viral replication and infectiousness potential. Additionally, a broad analysis of available SARS-CoV-2 genomes was also considered in the search for highly conserved non-mutated regions able to be targeted, without chances of having some virus variants capable of escaping from the selected siRNAs. Indeed, the chosen targets are present in all variants of interest that have already been sequenced, expanding the prospects for sustained successful clinical use. Other researchers also selected the same genes (1a and 1ab) as the best targets for siRNAs [28], although additional SARS-CoV-2 genes were also silenced by siRNA, achieving good results both in cell lines and in animal models [25]. Recently, siRNA treatment against SARS-CoV-2 was authorized for a clinical trial in Europe [29] because it achieved promising results in preclinical models [30].

After choosing the potential targets, a very rigorous in silico search for similar sequences in both humans and nonhuman genomic banks was performed, warranting a virtual “zero risk” for off-target events. This type of search is crucial for ensuring the reliability of our proposed targets, which are highly specific, and the presence of an anti-escape siRNA makes the proposed targets even more robust.

Compared to other siRNA experiments against SARS-CoV-2 infections carried out in nonhuman cells [31] and in transformed human cells [32], the developed nontransformed thyroid human cell model presents an important advantage, since it represents a SARS-CoV-2-susceptible normal human cell, allowing the investigation of the safety of the proposed treatment for normal human cells. In particular, off-target effects harming the normal human genome could be explored in an adequate model. Additionally, the effects of the infection and the potential protective benefits of the treatment could be investigated in a reliable model compared to the usually applied nonhuman and/or transformed cell models [33].

The proposed siRNAs were designed in-house to guarantee the agreed upon priorities, including high specificity, an almost null possibility of harming human RNAs, and not matching to any other organism’s genome, in addition to the conventional requirements for stable and efficient siRNA construction protocols.

Another improvement of the suggested strategy of combined siRNAs was related to choosing the best virus strand’s targets for both replication and escape sequences.

SARS-CoV-2 is a positive single-strand RNA virus, and after gaining access to human cells through ACE2 (angiotensin-converting enzyme II) cell receptors, the viral replication process encompasses a few steps that are critical for selecting the best targets, as briefly described [32].

The positive RNA strand uses the cellular machinery to transcribe two of its main genes, known as ORF1a and ORF1ab, but does not duplicate the whole virus genome. The peptides resulting from this human machinery-assisted process form the virus replicase that allows virus self-replication [34].

This replicase provides duplication of the whole positive strand, resulting in a negative strand. The negative strand is the generator of the new positive strands and finally allows the multiplication of the virus inside the human cell, keeping the infection active, since lots of new viruses will leave that cell to infect other host’s cells. The negative strand provides positive strand “products” by two different mechanisms: a negative strand is duplicated in a whole positive strand, the genomic replication, and a subgenomic replication process, where fragments of the positive strand, including specific structural genes, but not ORF1a and 1ab, are generated [35].

Looking at this rich scenario, many possibilities for addressing viral sequences can be realized, and the best targets for siRNA treatment remain under debate. However, targets that are durable in a virus that undergoes so many variations becomes challenging, as it is important to maintain efficiency in different viral variants.

Our choices were influenced by some concepts and conveniences. We decided to focus on the ORF1a/1ab region of the positive strand, since transcription of these regions is the first step for virus persistence in human cells and is absolutely necessary to produce the virus replicase, which is indispensable for sustainable infection [26].

In contrast to most previous attempts to control SARS-CoV-2 infection using a unique siRNA [25,33], combinations of siRNA were selected. Initially, four siRNAs were specifically designed to bind complementary sequences found exclusively in the SARS-CoV-2 genome and presented in every clinically relevant published SARS-CoV-2 genome, including variants α, β, γ, and Omicron. Impeding virus replication because of the inactivation of ORF1a/1ab should control the infection and, actually, the proposed siRNAs were extremely effective, reducing viral loads both individually and in different combinations, as demonstrated.

Although provoking extraordinary decreases in viral load and allowing cellular recovery after infection, a few days after the initial siRNA treatment, a subtle increase in virus production seemed to occur, nevertheless in a small fraction, compared to nontreated controls.

A hypothesis of a virus’s escape mechanism was considered, since extra doses of siRNAs failed to revert this phenomenon, and many viruses, including coronaviruses, have the capacity to escape siRNA attacks, as demonstrated in depth [12]. Accordingly, the virus might escape from siRNA by producing siRNA suppressors, so after being hit by the first dose of siRNAs, it counterattacks by increasing the production of siRNA suppressor sequences, making the new siRNA doses much less effective. Among putative virus escape strategies, the possibility of preventing the RNA-induced silencing complex (RISC), the effective complex containing the siRNAs that bind and inactivate the target sequences function by the production of specific sequences has been discussed.

Findings on genetic changes for SARS-CoV-2 indicate a high rate of variance in different areas of the genome; to exemplify this occurrence, we can cite studies showing a genome-wide nucleotide mutation rate of at least 0.67 substitutions per site annually, which is higher than other viruses such as Influenza [26,31].

An extensive in silico investigation permitted the conception and construction of an extra siRNA molecule designed against this putative anti-siRNA sequence, the incorporation of which in the siRNA combination blocked the assumed escape mechanism. Nevertheless, other escape mechanisms have been reported, such as modifications in ribosomes of infected cells allowing translation of temporary mutations [2,36]. The strategy of adding siRNA-V, the “anti-siRNA suppressor bullet”, seemed to be correct, since it actually counterattacked the putative RISC-based virus escape strategy, keeping the virus load much lower even after some days of treatment.

Although siRNA-V is designed to block viral replication, it must be considered that it was designed to target the nucleocapsid (N) protein region of SARS-CoV-2 and, in doing so, target an important factor for infection. viral replication, since this siRNA negatively regulates the N protein and reduces the packaging of the viral genome.

The Protein N is believed to perform multiple functions, including the formation of a helical ribonucleoprotein (RNP) complex during packaging of the RNA genome, replication, and regulation of viral RNA synthesis, transcription, and regulation of infected cell metabolism. The N protein safeguards the integrity and stability of the virus RNA by preserving it within the virus. RNA is wrapped in a long helical structure by N proteins [37,38].

Another strategic decision was that, instead of waiting for the supposed siRNA suppressor escape mechanism, the anti-escape siRNA was preliminarily launched at the onset of treatment as chess move anticipation.

Importantly, the choice for targeting the anti-siRNA sequence needed to be at a subgenomic region, since the in silico prediction pointed to a region away from ORF1ab. Considering that many positive-strand subgenomic fragments are produced, if the positive-strand target was selected, a tremendous reduction in efficacy should be anticipated due to the great number of targets to be inactivated. Instead of targeting those multiple fragments, our choice fell at the negative strand, the one that originates all positive subgenomic fragments.

Every discussed hypothesis was tested in an also innovative model: a non-transformed human cell, susceptible to SARS-CoV-2 infection. The results strongly demonstrated the capacity of controlling viral load, mainly, using the siRNAs combinations. The addition of an anti-escape siRNA improved the duration of infection control and favored normal cells recovery.

Since the selected siRNAs (protected by patents) showed immense potential in consistently reducing viral load without harming human cells, and considering the specificity of the molecular therapy, constructed to target conserved sequences present in every clinically relevant variant of the SARS-CoV-2, this strategy surges as a new important weapon to face COVID-19.

## 5. Conclusions

The combination of siRNAs I, III and V, addressing both virus replication and the siRNA suppressor escape mechanism, seems to be the best formulation for clinical investigation due to safety, efficiency, and prolonged sustained activity, imposing a checkmate to SARS-CoV-2.

These extraordinary preliminary results open the path to investigating this strategy in animal models and, subsequently, in human clinical trials, aiming to provide the potential benefits of this innovative approach against SARS-CoV-2 and COVID-19 and even paving the way for siRNA therapy applicability in several other infectious and chronic diseases.

## Figures and Tables

**Figure 1 genes-13-02147-f001:**
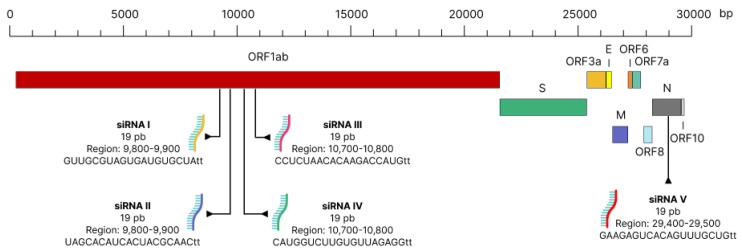
Schematic demonstrating the site of action of siRNA in the SARS-CoV-2 genome.

**Figure 2 genes-13-02147-f002:**
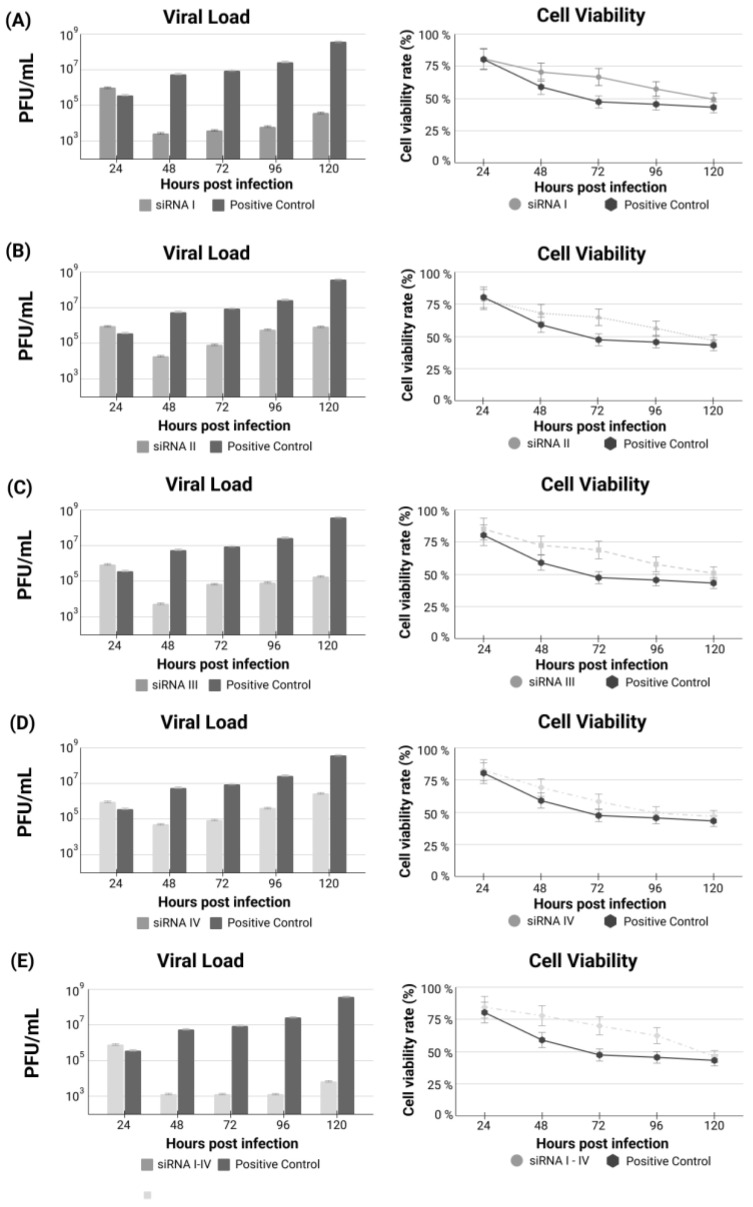
(**A**–**D**) Treatment with each siRNA individually (I to IV) after 24 h of SARS-CoV-2 infection. (**E**) Analysis of viral load and cell viability after treatment using the cocktail containing the combination of siRNAs I to IV.

**Figure 3 genes-13-02147-f003:**
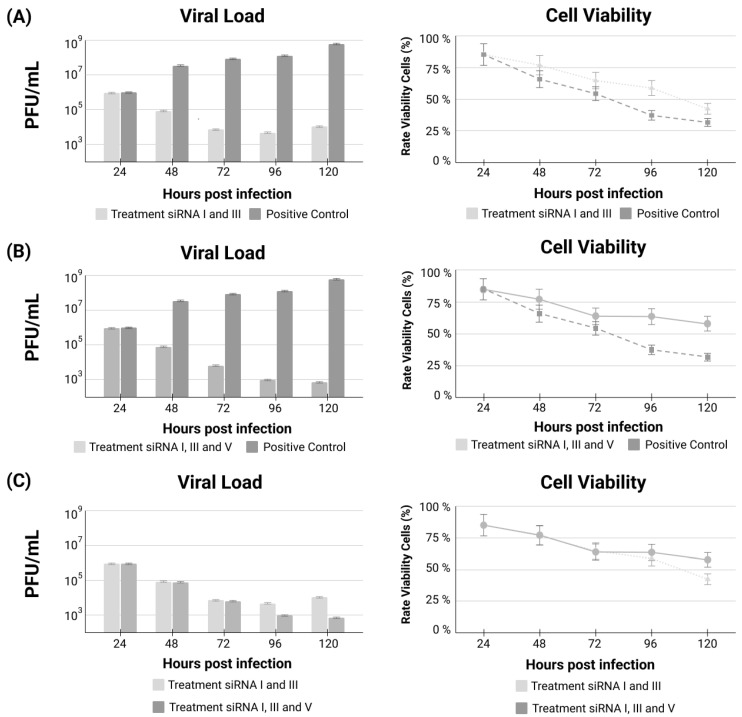
Viral load and cell viability comparing untreated cells versus both the combination of siRNA-I plus siRNA-III treatment (**A**); the combination of siRNA-I plus siRNA-III plus the addition of the anti-escape siRNA-V treatment (**B**); and the comparison between the two treatment combinations (**C**).

## Data Availability

The IMR-90 and VERO E6 cell lines was obtained from Edivaldo Herculano Correa de Oliveira at the Evandro Chagas Institute.

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
