# Peer review of "Anticipating the Next Chess Move: Blocking SARS-CoV-2 Replication and Simultaneously Disarming Viral Escape Mechanisms"

_genes, 2022, doi:10.3390/genes13112147_

Round 1

Reviewer 1 Report

General comments

This work uses siRNAs to treat SARS-CoV-2 infection in cultured human thyroid cells. To identify siRNA combinations that work optimally, the authors measured and monitored viral load, cell viability and caspase expression over 120 hours. A combination of three siRNAs (I, III, V) was found to reduce viral load and increase cell viability, providing a promising candidate for future in vivo studies. A weakness of the paper is that the design, selection and targets of siRNAs are only vaguely discussed; the siRNA sequences in the study are not presented. This prevents a proper assessment of the work's impact.

Specific comments

1. In Materials and Methods, the design, selection and targets of siRNAs should be elaborated (like in Ref. 26).

2. Several studies using siRNAs to treat SARS-CoV-2 have appeared in the last two years. These should be reviewed in the Introduction, and the work's contributions should be discussed in that context. For example, are the siRNA sequences and targets similar to those found in other studies?

3. Figure 2 and pages 9-10 suggest that siRNA-V is an anti-escape siRNA. To validate this claim, the viral/human siRNAs inhibited by the designed siRNA-V should be identified.

4. Figure 1, % cell viability at 120h is about the same with or without siRNA treatment. How can this be explained? The different effects of siRNAs need to be discussed based their targets.

Other comments

Page 2, "...no drug was found to present or support an efficient and specific treatment." This should be updated for 2022 (Paxlovid and therapeutic antibodies are routine, effective treatments).

Page 9, RISC is RNA-induced silencing complex. 

Page 9, 6,677x10^-4 can be more simply written as 0.67.

Figure 2, should siRNA IV be siRNA V? 

Reviewer 2 Report

In this study, Casseb et al. proposed siRNA combination therapy to reduce the viral load as well as the treatment to allow cellular recovery which might be a potential therapy to treat COVID-19.

Authors can find the comments and suggestions below

-This perspective seems to be interesting to the audience but it does need revisions to make it better with more logical outcomes and comparison with other studies as well as the language to make it more understandable to the readers.

- In abstract change “The proposed siRNA combination deeply reduced the viral load throughout the experiment and allowed cellular recovery, thus representing a potential innovation, to be considered as an additional weapon for therapy of COVID-19 and even other infectious diseases” to “The proposed siRNA combination could reduce the viral load and allow the cellular recovery, that presenting a potential innovation, to be considered as an additional strategy to counter or cope COVID-19.

-Be specific to siRNA in the introduction section and its potential implication. Moreover, the introduction part is too long, make it short.

- Author has done blocking SARS-CoV-2 replication and simultaneously disarming viral escape mechanisms but what about the recombination strategies used by the SARS-CoV2 earlier been reported that SAR-CoV2 a recombinant virus and recombination strategies are well established in CoVs (10.3390/pathogens9030240).

- Author could also see and cite these articles https://doi.org/10.56770/jcp2021525, https://doi.org/10.1038/s41392-022-00996-y, https://doi.org/10.1038/s41577-021-00578-z,

Round 2

Reviewer 1 Report

There a couple more corrections to consider.

Figure 1, change pb to bp; region of siRNA V is incorrectly labeled?

Figure 3B shows treatment with siRNA V but no labels in the figure are found.

In Discussion, I find the suggestion that siRNA V acts as a kind of anti-siRNA suppressor to be not very convincing. Since siRNA V targets the most abundant N transcripts, it downregulates N proteins and reduces packaging of the viral genome. This is a more plausible explanation of increased cell viability with siRNA V treatment.

Author Response

Q1 - Figure 1, change pb to bp; region of siRNA V is incorrectly labeled?

Thanks for the information the correction was made in the figure

Q2 - Figure 3B shows treatment with siRNA V but no labels in the figure are found.

Thanks for the information the correction was made in the figure

Q3 - In Discussion, I find the suggestion that siRNA V acts as a kind of anti-siRNA suppressor to be not very convincing. Since siRNA V targets the most abundant N transcripts, it downregulates N proteins and reduces packaging of the viral genome. This is a more plausible explanation of increased cell viability with siRNA V treatment.

It is worth noting that our siRNA-V, "anti-siRNA suppressor bullet," was designed to target the region of the nucleocapsid protein (N) of SARS-CoV-2 and, in doing so, to reach an important factor for viral replication, as it downregulates the N protein and reduces the packaging of the viral genome.

We appreciate your contribution, and in order to clarify the significance of siRNA-V, we have added the following to our manuscript:

"Although siRNA-V is designed to block viral replication, it must be considered that it was designed to target the nucleocapsid (N) protein region of SARS-CoV-2 and, in doing so, target an important factor for infection. viral replication, since this siRNA negatively regulates the N protein and reduces the packaging of the viral genome.

The Protein N is believed to perform multiple functions, including the formation of a helical ribonucleoprotein (RNP) complex during packaging of the RNA genome, replication, and regulation of viral RNA synthesis, transcription, and regulation of infected cell metabolism. The N protein safeguards the integrity and stability of the virus RNA by preserving it within the virus. RNA is wrapped in a long helical structure by N proteins."